# Lithium Vanadium Oxide/Graphene Composite as a Promising Anode for Lithium-Ion Batteries

**DOI:** 10.3390/nano13010043

**Published:** 2022-12-22

**Authors:** Leichao Meng, Jianhong Peng, Yi Zhang, Yongfu Cui, Lingyun An, Peng Chen, Fan Zhang

**Affiliations:** 1Qinghai Provincial Key Laboratory of Nanomaterials and Technology, School of Physics and Electronic Information Engineering, Qinghai Minzu University, Xining 810007, China; 2School of Energy Sciences and Engineering, Nanjing Tech University, Nanjing 211816, China; 3School of Materials and Chemical Engineering, Tongren University, Tongren 554300, China; 4School of Materials Science and Engineering, East China Jiaotong University, Nanchang 330013, China

**Keywords:** lithium vanadium oxide, graphene, anode, lithium-ion batteries

## Abstract

Lithium vanadium oxide (Li_3_VO_4_, LVO) is a promising anode material for lithium-ion batteries (LIBs) due to its high theoretical capacity (394 mAh g^−1^) and safe working potential (0.5–1.0 V vs. Li^+^/Li). However, its electrical conductivity is low which leads to poor electrochemical performance. Graphene (GN) shows excellent electrical conductivity and high specific surface area, holding great promise in improving the electrochemical performance of electrode materials for LIBs. In this paper, LVO was prepared by different methods. SEM results showed the obtained LVO by sol-gel method possesses uniform nanoparticle morphology. Next, LVO/GN composite was synthesized by sol-gel method. The flexible GN could improve the distribution of LVO, forming a high conductive network. Thus, the LVO/GN composite showed outstanding cycling performance and rate performance. The LVO/GN composite can provide a high initial capacity of 350.2 mAh g^−1^ at 0.5 C. After 200 cycles, the capacity of LVO/GN composite remains 86.8%. When the current density increased from 0.2 C to 2 C, the capacity of LVO/GN composite only reduced from 360.4 mAh g^−1^ to 250.4 mAh g^−1^, demonstrating an excellent performance rate.

## 1. Introduction

In recent decades, traditional fossil energy (coal, oil, natural gas) has been widely exploited and utilized, which bring serve negative impacts on the environment such as air pollution, climate warming and other issues. Moreover, fossil energy sources are not renewable. Therefore, it is essential to developing environmentally friendly, renewable and sustainable green energy (wind, solar, etc.). However, the green energy is not continuous. Energy storage devices play an important role in utilizing green energy [1,2,3,4,5,6]. Among them, lithium-ion batteries (LIBs) show many advantages such as high energy density, high operating voltage, no memory effect, and being environmentally friendly [7,8,9,10]. Thus, LIBs have been widely used in various fields, including portable electronic devices, hybrid vehicles, pure electrical vehicles (EVs), and aerospace.

With the rapid development of EVs, there is an increasing demand for EVs’ LIBs electrochemical performance. In general, the electrochemical performance of batteries is related to the electrode active materials’ intrinsic properties such as theoretical capacity, electrical conductivity, and volume change during cycling [11,12,13]. At present, the commercial LIBs’ anode active material is graphite which is stable during cycling and low cost. However, its theoretical capacity is low (372 mAh g^−1^). In addition, the chemical kinetics of graphite is slow. Moreover, the Li^+^ intercalation potential of graphite is low (0.1 V), which leads to lithium dendrites easily forming during charge and discharge, resulting in irreversible lithium loss and battery safety issues. Therefore, the development of novel anode materials with high-capacity, safe Li^+^ intercalation potential, and long cycle life such as silicon [14], metal oxide [15], transition metal oxides [16] have attracted more and more attention.

Among various anode materials, lithium vanadium oxide (Li_3_VO_4_, LVO) is a promising anode material for LIBs due to its high theoretical capacity (394 mAh g^−1^) and safe working potential (0.5–1.0 V vs. Li^+^/Li) [17]. However, its electrical conductivity is low which leads to poor electrochemical performance. Combining LVO with high conductive carbon material is an effective method to enhance LVO electrochemical performance. Zhai et al. prepared carbon-coated nanosized LVO (LVO-BMC) through ball-milling route followed by CVD technique [18]. The LVO-BMC showed an initial discharge capacity of 396 mAh g^−1^ at 20 mA g^−1^, and maintained 245 mAh g^−1^ after 50 cycles with an improved capacity retention of 61.9%. Zhang et al. synthesized carbon encapsulated LVO with core-shell nanostructure in which the carbon was uniformly coated on the LVO [19]. Thus, LVO/C showed enhanced cycling stability (maintain 401 mAh g^−1^ at 0.1 C after 50 cycles). Zhao et al. fixed LVO nanoparticles in graphitized porous carbon (HP-LVO/C) [20]. The HP-LVO/C showed significantly improved cycle stability (kept a high-capacity of 381 mAh g^−1^ at 0.2 A g^−1^ after 100 cycles) because the hierarchically porous carbon can provide fast ion transport path and release the volume change of LVO during cycling.

In this work, we prepared LVO by different methods (solid-phase method, hydrothermal method and sol-gel method). The morphology of LVO was studied to confirm the suitable synthesis method. Next, LVO/GN composite was synthesized by sol-gel method. The flexible GN could inhibit the volume expansion of LVO and enhance the electrical conductivity of the LVO/GN composite. As a result, the LVO/GN composite showed a high initial capacity of 350.2 mAh g^−1^ at 0.5 C with a high-capacity retention of 86.8% after 200 cycles, which was higher than that of other researchers’ results [18,19,20]. When the current density increased from 0.2 C to 2 C, the capacity of LVO/GN composite only reduced from 360.4 mAh g^−1^ to 250.4 mAh g^−1^, demonstrating excellent rate performance. Our LVO/GN composite provided a promising option for high-performance anode materials for LIBs.

## 2. Experimental

### 2.1. Preparation of LVO

The LVO was synthesized by solid-phase method, hydrothermal method and sol-gel method. In solid-state processes, Li_2_CO_3_ and V_2_O_5_ powders with a mole ratio of 3:1 were ground for 0.5 h. The mixture was put into a muffle furnace and kept at 600 °C for 10 h in air with a heating rate of 5 °C min^−1^. The obtained white sample was LVO (named as S-LVO). For hydrothermal method, LiOH and V_2_O_5_ with a mole ratio of 6:1 were mixed in 100 mL deionized water and then poured into a Teflon-lined stainless-steel autoclave at 180 ℃ for 20 h in an oven. The obtained sample was LVO (named as H-LVO). The sol-gel processes were as follows: Li_2_CO_3_ and V_2_O_5_ powders with a mole ratio of 3:1 were mixed in 100 mL deionized water. The mixture was dried in an oven at 90 ℃ for 12 h to obtain the precursor powder. Next, the precursor powder was calcined in a tubular furnace at 600 ℃ for 10 h under Ar. The obtained sample was LVO (named as G-LVO).

### 2.2. Preparation of LVO/GN Composite

The LVO/GN composite was synthesized by sol-gel method. In this case, 200 mg graphene oxide was added in 500 mL deionized water. Next, 30 mmol Li_2_CO_3_ and 10 mmol V_2_O_5_ were mixed in graphene oxide solution. The mixture was dried in an oven at 90 ℃ for 12 h to obtain the precursor powder. Next, the precursor powder was calcined in a tubular furnace at 600 ℃ for 10 h under Ar. The obtained sample was LVO/GN composite. The schematic illustration of LVO/GN composite synthesis is shown in Figure 1.

### 2.3. Characterization

The structure of LVO and LVO/GN composite was characterized by X-ray diffraction (XRD, Rigaku MiniFlexll, Rigaku Corporation, Tokyo, Japan). The morphologies of LVO and LVO/GN composite was characterized by scanning electron microscope (SEM, Zeiss Supra 55 and Phenom Prox, Carl Zeiss, Oberkochen, Germany). Thermal gravimetric analysis was performed by a thermal analyzer (TGA, TGA/SDTA 851, Mettler Toledo, Greifensee, Switzerland).

### 2.4. Electrochemical Measurements

In order to prepare anode, we mixed 80% LVO or LVO/GN composite, 10% acetylene black and 10% polyvinylidene fluoride (PVDF) in N-methylpyrrolidone (NMP) to form a slurry which was then coated on copper foil and dried at 100 °C for 12 h. The mass loading of active material is about 1.5 mg cm^−2^. We assembled it into a CR2025 coin cell in the glove box filled with high-purity Ar, with lithium foil as counter electrode, polypropylene film (Celgard 2400) as separator and 1.0 M LiPF_6_ in ethylene carbonate (EC)/dimethyl carbonate (DMC)/carbon Ethyl methyl acetate (EMC) (in a volume ratio of 1:1:1) with 2% vinylene carbonate (VC) additives as electrolyte. Cyclic voltammetry (CV) and electrochemical impedance spectroscopy (EIS) tests were performed using a CHI660D electrochemical workstation. The cycling performance was tested by a LAND CT2001A battery tester (Wuhan, China).

## 3. Results and Discussion

Figure 2a was the XRD patterns of LVO samples synthesized by solid-phase method, hydrothermal method and sol-gel method. All LVO samples showed peaks at 2θ of 16.1°, 21.6°, 22.7°, 24.1°, 28.0°, 32.9°, 36.2°, 36.4°, 37.5°, 40.6°, 43.1°, 47.9°, 50.4°, 56.7°, 57.5°, 58.4°, 67.1°, and 71.6° which were, respectively, attributed to the (100), (110), (011), (101), (111), (200), (210), (002), (201), (112), (211), (221), (202), (230), (300), (320), (203), and (322) diffraction planes of spinel Li_3_VO_4_ (JCPDS No. 38-1247) [21,22]. The crystallite size can be obtained by Debye Scherrer equation *d* = *K λ*/*β* cos*θ* in which *d* = average crystallite size, *K* = the sharp factor, *λ* = the X-ray wave length, *β* = the reflection width and *θ* = the diffraction angle [23]. The crystallite sizes of S-LVO, H-LVO, G-LVO calculated from the XRD data were about 5 μm, 3 μm, 1 μm. For the XRD diffraction pattern of H-LVO, there were another weak diffraction peaks at about 30°. These peaks are attributed to the (002) and (202) diffraction planes of the Li_2_CO_3_ (JCPDS No. 36-0787) [24]. XRD results show that the purity of LVO sample by hydrothermal method is not high. Hydrothermal synthesis refers to the synthesis through chemical reactions in an aqueous solution above the boiling point of water. The reaction temperature of hydrothermal method is relatively low. Thus, the sample by hydrothermal method is not high purified. The morphologies of three LVO samples were studied to confirm the best synthesis method. Figure 2b is the SEM image of S-LVO (by solid-phase method). It can be seen that S-LVO particle is relatively large (2–10 μm). In addition, the particle size is various. As shown in Figure 2c, H-LVO particle is also big. However, the particle size is uniform. From Figure 2d, the LVO sample by sol-gel method showed a relatively small and uniform particle size (1 μm). From the morphologies of three LVO samples, the sol-gel method is suitable for preparing LVO nanoparticles.

According to the analysis above, LVO/GN composite was synthesized by sol-gel method. XRD was carried out to analyze the structure of LVO/GN composite. As shown in Figure 3a, LVO/GN composite showed peaks at 2θ of 16.2°, 21.7°, 22.8°, 24.2°, 28.1°, 33.0°, 36.3°, 36.5°, 37.6°, 40.7°, 43.2°, 47.9°, 50.5°, 56.8°, 57.6°, 58.5°, 67.2°, and 71.7° which is similar to that of LVO. The GN’s peaks were not observed in XRD pattern of LVO/GN composite, which is due to low GN content in LVO/GN composite. In order to analyze the content of GN, TG experiment of LVO/GN composite was conducted (Figure 3b). From room temperature to 350 °C, the weight loss is 5% which is due to the evaporation of free water on the sample’s surface [25,26,27]. From 350 °C to 700 °C, the weight loss is 11.2% which is the reaction of GN to CO_2_. When the temperature reached at 700 °C, the weight kept stable. The final product is only LVO. Therefore, the content of GN in the LVO/GN composite is 11.2 wt.%. Figure 3c is the SEM image of LVO/GN composite. Figure 3c showed most LVO nanoparticles uniformly dispersed on GN. Figure 3d is the TEM image of LVO/GN composite. Figure 3d confirmed the uniform distribution of LVO particles on GN. Moreover, the LVO particle size decreased to below 500 nm due to GN inhibiting LVO particle growth. The uniformly dispersed conductive structure can improve the electrical conductivity of LVO/GN composite and buffer the volume expansion of LVO during cycling.

In order to investigate the electrochemical performance of LVO/GN composite, CV experiment was carried out. Figure 4a showed the initial three-cycle CV curves of LVO in the voltage range of 0–3.0 V at a scan rate of 0.2 mV s^−1^. The green line is the first cycle. In addition, the red and blue lines are the 2nd and 3rd cycle, respectively. In the first cycle CV curve, three reduction peaks appeared at 0.75 V, 0.62 V and 0.25 V. The peak at 0.25 V could be attributed to the formation of solid electrolyte interface (SEI) film [28,29,30,31,32]. The peaks at 0.75 V and 0.62 V corresponded to Li^+^ intercalation into the LVO. The oxidation peak at 1.35 V indicated that Li^+^ deintercalated from LVO [33,34]. In the subsequent cycles, the reduction peaks changed from 0.75 V and 0.62 V to 0.85 V and 0.72 V while the oxidation peak changed from 1.35 V to 1.32 V. Moreover, the peaks’ area increased, which can be attributed to the activation during lithiation and delithiation [35]. For LVO/GN composite, CV curves were similar to that of pure LVO. However, the current of LVO/GN composite was obviously higher than that of LVO (Figure 4b), implying fast chemical kinetics for LVO/GN composite. Figure 4c showed the galvanostatic charge-discharge curves of the first cycle for LVO/GN electrode at 0.2 C. The galvanostatic charge-discharge curves showed a voltage plateau at 0.85 and 1.35 V, which was consistent with the CV results.

Figure 4d was the cycling performance of LVO and LVO/GN composite. The LVO/GN composite exhibited improved cycling performance. At 0.5 C, the initial reversible capacity of LVO/GN composite was as high as 350.2 mAh g^−1^. After 200 cycles, the capacity was 303.9 mAh g^−1^ with a high-capacity retention of 86.8%. For LVO, the capacity fast decreased during the first 15 cycles and slightly increased and then kept stable in the subsequent cycles. This phenomenon was observed in other vanadium oxide [36]. The rapid capacity decrease is resulted from the volume expansion and formation of SEI layer. The capacity’s slight increase might then be attributed to the increased crystallinity of active materials and gradual activation progress. The subsequent capacity kept stable due to the complication of activation progress. After 200 cycles, the capacity of LVO was 248.1 mAh g^−1^, and the capacity retention was as low as 71.8%. The capacity was calculated by the total mass of active material in our study. For LVO/GN composite, GN also contributed capacity. At 0.5 C, GN showed a stable capacity of 282.6 mAh g^−1^ [37]. We can obtain the capacity of G-LVO in G-LVO/GN composite by the equation: Q_G-LVO/GN_ (303.9 mAh g^−1^) = *W*_G-LVO_ (88.8 wt%) × Q_G-LVO_ + *W*_GN_ (11.2 wt%) × Q_GN_ (282.6 mAh g^−1^). The G-LVO in G-LVO/GN composite delivered capacity of 306.8 mAh g^−1^ at 0.5 C which is higher than that of pure LVO (248.8 mAh g^−1^). Moreover, LVO/GN composite showed good rate performance (Figure 4e). At 0.2 C, 0.5 C, 1.0 C, 2.0 C, 5.0 C, and 10.0 C, LVO/GN composite showed high discharge capacities of 363.2 mAh g^−1^, 321.3 mAh g^−1^, 301.9 mAh g^−1^, 251.8 mAh g^−1^, 180.5 mAh g^−1^, 100.9 mAh g^−1^, respectively. The charge capacities of LVO/GN composite at 0.2 C, 0.5 C, 1.0 C, 2.0 C, 5.0 C, and 10.0 C were 360.4 mAh g^−1^, 319.8 mAh g^−1^, 300.3 mAh g^−1^, 250.4 mAh g^−1^, 179.6 mAh g^−1^, 100.3 mAh g^−1^, respectively. Under the same rate, the discharge capacities of LVO were only 252.3 mAh g^−1^, 220.6 mAh g^−1^, 180.9 mAh g^−1^, 110.6 mAh g^−1^, 49.9 mAh g^−1^, 40.2 mAh g^−1^ while the charge capacities of LVO were only 249.7 mAh g^−1^, 219.8 mAh g^−1^, 180.3 mAh g^−1^, 110.2 mAh g^−1^, 49.8 mAh g^−1^, 40.2 mAh g^−1^, respectively. The galvanostatic charge-discharge curves of the LVO/GN composite at various rates confirm the good rate performance for LVO/GN composite (Figure 4f). The improved electrochemical performance of LVO/GN composite can be attributed to the positive impact of GN and LVO. Compared with commercial graphite anode, LVO/GN composite showed a higher capacity because the theoretical capacity of LVO is higher than that of graphite. Moreover, LVO/GN composite showed better cycling performance than pure LVO. The high mechanical performance of GN could significantly buffer the volume change of LVO during cycling processes, effectively suppressing the electrode pulverization, consequently ensuring the high cyclic stability. GN with high electrical conductivity can increase the electrical conductivity of LVO through offering conductive highways [38,39]. Moreover, small LVO nanoparticles were uniformly distributed on GN to form a multi-dimensional integrated structure, building a comprehensive electron/ion transport network and stable structure, which is beneficial for the cyclic performance [40,41,42].

In order to characterize the positive effect of GN on LVO, EIS analysis for LVO and LVO/GN composite was carried out. Figure 5a showed the EIS spectrum of LVO and LVO/GN composite. In the EIS spectrum, the high-frequency region is related to the charge transfer process which reveals the characteristics of electrochemical reaction resistance while the low-frequency region is related to the electrical conductivity of electrode material [43,44,45]. The equivalent circuit model of EIS spectrum is shown in Figure 5b, where R is the interface resistance, R_ct_ is the charge transfer resistance, CPE is the capacitance, and W is the Warburg impedance. The R_ct_ of LVO/GN composite is smaller than that of LVO electrode, indicating that GN effectively enhances the electrical conductivity of LVO and improves the reaction kinetics.

## 4. Conclusions

In conclusion, a high-performance anode material for LIBs was obtained by successfully anchoring LVO nanoparticles uniformly on the GN through a simple sol-gel method. The GN not only improves the ion/electron transport kinetics but also acts as a buffer matrix to effectively alleviate the volume change of LVO during the continuous Li^+^ intercalation/delithiation processes. In addition, LVO nanoparticles were inhibited by GN, which can alleviate the LVO particle growth. As a result, the LVO/GN composite exhibited high reversible capacity (350.2 mAh g^−1^ at 0.5 C), good cyclic performance (303.9 mAh g^−1^ after 200 cycles at 0.5 C with a high-capacity retention of 86.8%), and excellent rate capability (100.3 mAh g^−1^ at 100 C). Therefore, the LVO/GN composite is a promising anode material for high-performance LIBs. In addition, the rational design in this work provides a guide for decorating various nanoparticles on the GN, and these materials can be widely used in energy storage, catalysis and other fields.

## Figures and Tables

**Figure 1 nanomaterials-13-00043-f001:**
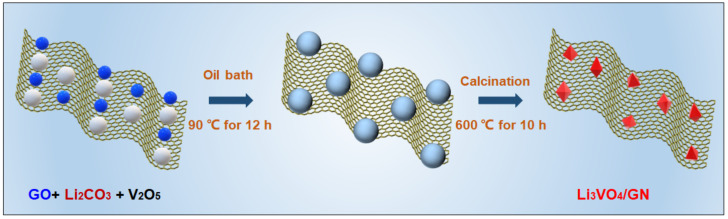
Schematic illustration of LVO/GN composite synthesis.

**Figure 2 nanomaterials-13-00043-f002:**
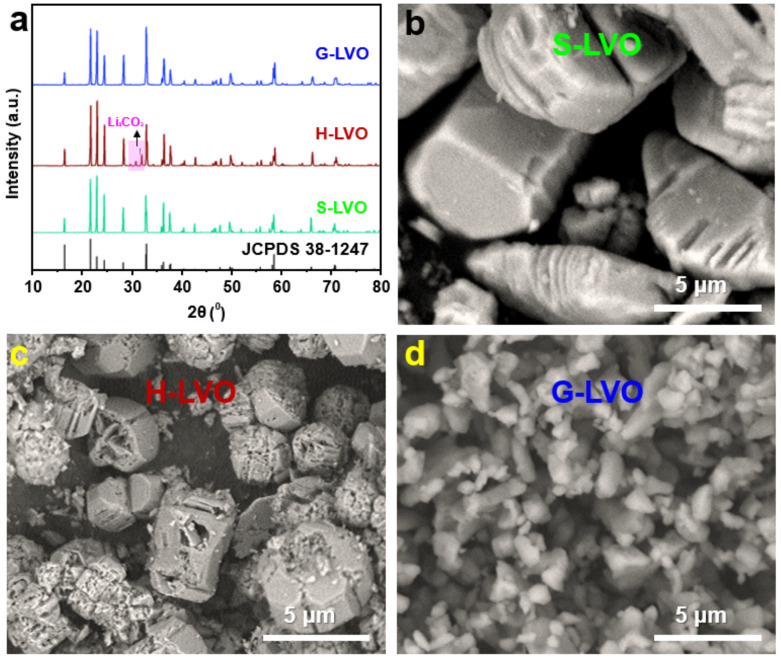
(**a**) XRD patterns of G-LVO, H-LVO, and S-LVO. (**b**) SEM image of S-LVO. (**c**) SEM image of H-LVO. (**d**) SEM image of G-LVO.

**Figure 3 nanomaterials-13-00043-f003:**
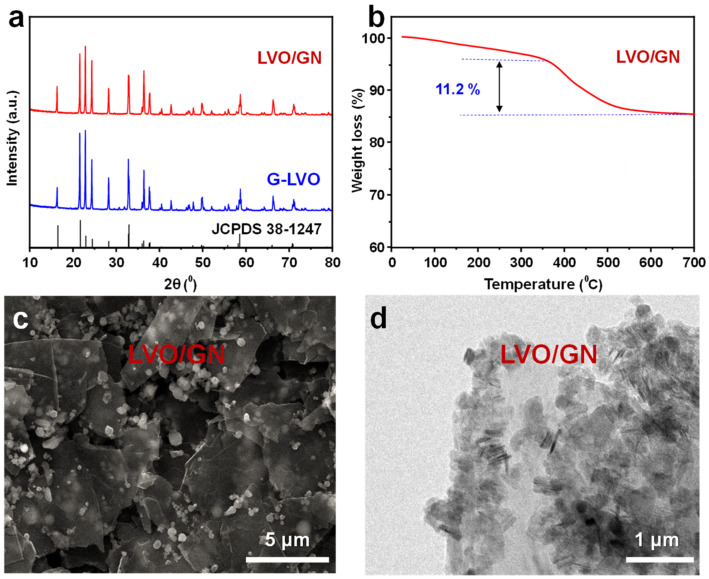
(**a**) XRD patterns of LVO and LVO/GN composite. (**b**) TGA curve of LVO/GN composite. (**c**) SEM image of LVO/GN. (**d**) TEM image of LVO/GN.

**Figure 4 nanomaterials-13-00043-f004:**
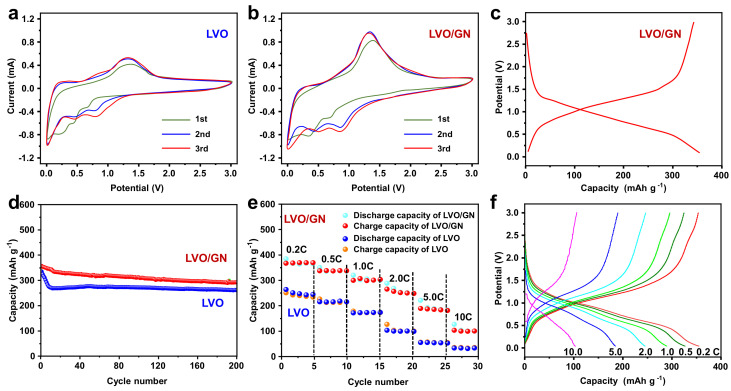
(**a**) CV curves of LVO electrode at a scan rate of 0.2 mV s^−1^. (**b**) CV curves of LVO/GN electrode at a scan rate of 0.2 mV s^−1^. (**c**) The charge/discharge curves of LVO/GN electrode at 0.2 C. (**d**) The cycling performance of LVO electrode and LVO/GN electrode at 0.5 C. (**e**) The rate performance of LVO electrode and LVO/GN electrode. (**f**) The charge/discharge curves of LVO/GN electrode at 0.2 C, 0.5 C, 1.0 C, 2.0 C, 5.0 C, 10.0 C. (1 C = 400 mA g^−1^).

**Figure 5 nanomaterials-13-00043-f005:**
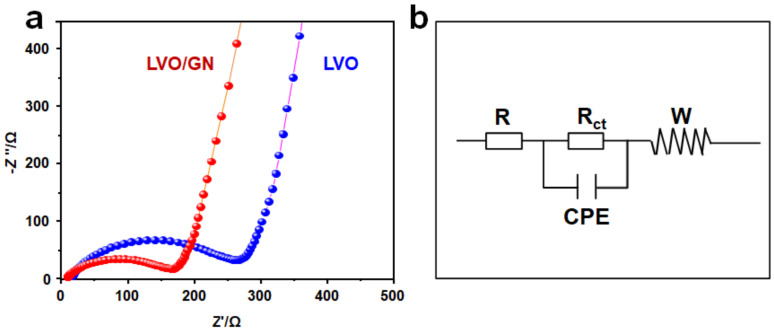
(**a**) Nyquist plots of LVO and LVO/GN. (**b**) The equivalent circuit model.

## Data Availability

Not applicable.

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
