# Peer review of "Lithium Vanadium Oxide/Graphene Composite as a Promising Anode for Lithium-Ion Batteries"

_nanomaterials, 2022, doi:10.3390/nano13010043_

Round 1

Reviewer 1 Report

The authors report the synthesis and characterization of a Li3VO4/GN composite to be used as a Li-ion anode material.  The experimental gravimetric capacity of this material is a bit more than that of graphite, while its potential is significantly higher.  The rate performance is clearly improved over that of graphite making this a useful addition to the literature.  It would have been better to include some comparison to graphite in the manuscript, however.

I recommend publication of this manuscript after major revisions as described below and after making the minor corrections I have annotated in the manuscript PDF itself.

In figure 2a, the diffraction pattern for H-LVO appears to have additional diffraction peaks near 30 degrees.  In addition the other two synthesized materials may have secondary phases.  Instead of simply listing Miller indices or angles, the authors need to perform Rietveld refinements to properly characterize the diffraction patterns, reporting lattice parameters, particle sizes, and amount of secondary phases.  This can be done with free software such as GSAS2 if they do not have commercial software with their diffractometer.  Such characterization is important as it can affect the capacity calculations.

In figures, 4a and 4b, the authors need to identify which CV cycles are which.  This can be done in the figure caption but it needs to be present.  In 4e there is no description of the orange and cyan points.  I presume they are for delithiation but this needs to be explained in the figure caption.  The caption for 4f duplicates that of 4e (see marked-up document).

The authors do not explain how the capacity for the composite G-LVO/GN is computed.  Is it only a function for the G-LVO or do they include the GN in the calculation?  This is important to explain clearly as there is 11.2% of GN in the composite.  From the description of the cell preparation in section 2.4, it appears as if the amount of LVO material in the composite is about 70% compared to 80% in the pure LVO cell due to the 11.2% GN in the composite.  The overall capacity of the composite cell could also have some contribution from the GN itself and that needs to be accounted for in order to determine if the effective capacity of the LVO itself is increased.  The rate capability is clearly improved.

The paper would be enhanced with a more thorough discussion of the significance of these results, including an analysis of the comparison to graphite.

Author Response

We thank the reviewer for the positive comments and detailed suggestions. Following the constructive suggestions, we revised our paper in a point-by-point manner. Please see the attachment.

Reviewer 2 Report

The presented research "Composite lithium vanadium oxide/graphene as a promising anode for lithium-ion batteries" demonstrates the possibility of synthesizing the composite material lithium vanadate - reduced graphene oxide (LVO-GN). The material exhibits good electrochemical performance in a cell with a lithium counter electrode. The morphology and composition of the synthesized material are described by scanning electron microscopy and X-ray phase analysis. Despite the good level of work, some moments should by clarified:

1.       In the introduction, the need to search for new electrode materials and improve existing ones are discussed. A couple of examples of the relevance of using Li3VO4 as an anode for metal-ion batteries are also given. However, there is practically no data on composite materials based on carbonaceous materials and Li3VO4. Since the manuscript describes a material based on reduced graphene oxide and Li3VO4, in particular, its electrochemical properties as an anode in a lithium-ion battery, it is advisable to compare the obtained parameters with data for materials with a similar composition, that were obtained earlier and described in the literature. What will make it possible to adequately evaluate obtained results

2.       The authors demonstrate the diffraction patterns of the obtained materials, but do not provide the crystallite sizes calculated from these data (based on Scherrer equation).

3.       Figure 3 shows the SEM images of graphene oxide containing material. The authors note a uniform distribution of Li3VO4 particles over the surface of graphene sheets. However, it is difficult to draw such a conclusion based on SEM images. Agglomerates of different sizes are randomly scattered over the graphene surface.

4.       Are there smaller vanadate particles (less than 500 nm) on the surface of reduced graphene oxide particles? Or, during synthesis, vanadate particles with a size of about 500 nm are formed from the precursor solution? It is necessary to conduct a study of the material by the method of transmission electron microscopy or provide elements distribution maps by the EDXS method. This will significantly enhance the experimental part of the manuscript.

5.       Figure 4 (a, b) shows the CV curves of LVO and LVO-GN. As far as I understand, the green line refers to the 1st cycle, and the red and blue lines to the 2nd and 3rd, respectively. How can you explain the higher area under the curves for the 2nd and 3rd cycles (which basically corresponds to the specific capacitance of the cell), while the specific capacitance values decrease from the 1st to the 3rd  cycle.

6.       How can one explain the dramatic drop the reversible capacity for LVO in the first 10-15 cycles (Figure 4d) with future stabilization up to 200 cycles?

Author Response

We thank the reviewer for taking the time to review our article and for providing positive comments and detailed suggestions. We are pleased that the reviewer recognized this work. Based on constructive suggestions, we revised our paper point by point. Please see the attachment.

Round 2

Reviewer 1 Report

The major points flagged in the first review have been addressed satisfactorily.  I would, however, usre the authors to learn to use GSAS2 and to perform Rietveld refinement on all their materials.  Weak diffraction peaks can be deceptive and correspond to a surprisingly large amount of impurity phase by weight.  There is a lot of information available in a diffraction pattern and it should be used.  It is also a very convincing way to show reviewers that they understand their material.

In addition, my suggestions annotated in the original manuscript have not been made.  I attach them below

1. on the first line of the introduction, "foil" is not the correct word and is likely supposed to be "fossil"

2. on the first line of section 2.1 I suggest "solid-state" as a more correct name father than "solid-phase"

3. "deintercalated" insted of "detercalated" in the highlighted paragraph below Figure 3

4. "interface" instead of "inretface" below Figure 4

5. "EIS spectrum is shown" (replace "was" with "is") just below Figure 4

I recommend publication with the minor typographical changes above.

Author Response

Thanks a lot for your suggestion. We have made some changes according to your comments. Please see the attachment.

Reviewer 2 Report

I want to thank the authors for detailed responses to the remarks and comments. The manuscript can be accepted for publication in present form.

Author Response

We thank the reviewer for taking the time to review our article. We are pleased that the reviewer recognized this work. Thank you.